# Imitation Bootstrapped Reinforcement Learning

## Abstract

Despite the considerable potential of reinforcement learning (RL), robotics control tasks predominantly rely on imitation learning (IL) owing to its better sample efficiency. However, given the high cost of collecting extensive demonstrations, RL is still appealing if it can utilize limited imitation data for efficient autonomous self-improvement. Existing RL methods that utilize demonstrations either initialize the replay buffer with demonstrations and oversample them during RL training, which does not benefit from the generalization potential of modern IL methods, or pretrain the RL policy with IL on the demonstrations, which requires additional mechanisms to prevent catastrophic forgetting during RL fine-tuning. We propose *imitation bootstrapped reinforcement learning* (IBRL), a novel framework that first trains an IL policy on a limited number of demonstrations and then uses it to propose alternative actions for both online exploration and target value bootstrapping. IBRL achieves SoTA performance and sample efficiency on 7 challenging sparse reward continuous control tasks in simulation while learning directly from pixels. As a highlight of our method, IBRL achieves $6.4\times$ higher success rate than RLPD, a strong method that combines the idea of oversampling demonstrations with modern RL improvements, under the budget of **10** demos and **100K** interactions in the challenging PickPlaceCan task in the Robomimic benchmark.

## 1 Introduction

Despite achieving remarkable performance in many simulation domains (Silver et al., 2017; Vinyals et al., 2019; Mirhoseini et al., 2021; FAIR et al., 2022), reinforcement learning (RL) has not been widely used in solving robotics and low level continuous control problems. The main challenges of applying RL to these continuous control problems are exploration and sample efficiency. In these settings, reward signals are often sparse by nature, and unlike learning in games where the sparse reward is often achievable within a maximum amount of steps, a randomly initialized neural policy may never finish a task, resulting in no signals for learning. Even for tasks with hand-engineered dense reward functions, RL may still need a large number of samples to converge, which hinders its adoption in the real world where massive parallel simulation is not available.

As a result, most learning-based robotics systems largely rely on imitation learning (IL) (Brohan et al., 2023) or offline RL (Kumar et al., 2022) with strong assumptions such as access to large specialized datasets. However, those methods come with their own challenges. Expert demonstrations are often expensive to collect requiring access to expert operators and domain knowledge (Mandlekar et al., 2021). In addition, policies learned from static datasets suffer from distribution shifts when deployed in slightly different environments. Given these challenges, online RL algorithms – when carefully integrated with IL – can still play a valuable role in efficiently learning robot policies.

Prior RL methods primarily utilize human demonstrations in two ways. The most straightforward one is to initialize the RL replay buffer with demonstrations and oversample those demonstrations during training (Vecerík et al., 2017). While this idea can be effective, it does not benefit from the potential generalization that IL policies may have gained beyond the limited demonstration data. The second way is to pretrain the RL policy with human data and then fine-tune it with RL while applying additional procedures to ensure that pretrained policy initialization does not get washed out quickly by the randomly initialized critics. For example, Haldar et al. (2022) use an adaptive regularization loss while Hansen et al. (2023a) also pretrain the critic and a world model on data

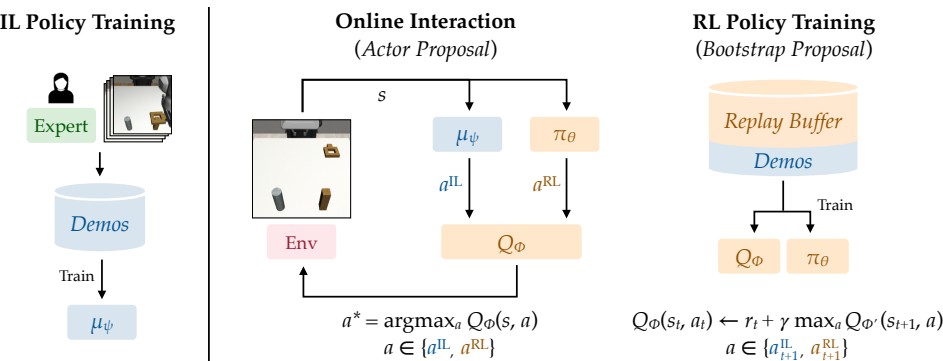

Figure 1: Overview. IBRL first trains an imitation learning policy and then uses it to propose additional actions for RL during both online interaction (actor proposal) and training phases (bootstrap proposal). IBRL selects the action with higher Q-values using the online and target Q-networks in the two phases respectively.

generated by the IL policy. The pretrained critic and the model-based approach are crucial to prevent the policy from collapsing, but this is at the cost of increased computational complexity. Despite efforts to prevent the policy from forgetting the knowledge extracted from the data, these methods may not fully utilize the potential of the IL policy since the same policy needs to digest two datasets of different distributions: the initial demonstrations and the data generated by RL. Additionally, these prior methods implicitly require the IL policy to have the same architecture as the RL policy, which limits the selection of the network architectures to use during pretraining.

Our goal is to design a sample efficient RL algorithm that can effectively utilize prior data and autonomously self-improve. The goal is for this policy to achieve better performance than using only IL on the prior data, and to mitigate deterioration caused by distribution shift. In this paper, we propose *Imitation Bootstrapped Reinforcement Learning* (IBRL), a novel method to combine IL and RL for sample efficient learning. IBRL first trains an imitation policy on the provided demonstrations, and then explicitly uses this IL policy in two phases to *bootstrap* standard RL training. First, both the IL policy and RL policy propose an action during the online interaction phase, and the agent executes the action by either policy that has a higher Q-value. Second, during the training phase of RL, the target for updating the Q-values bootstraps from the *better* action among the ones proposed by either the RL or the IL policies. Additionally, similar to prior work, the replay buffer for RL is pre-filled with the demonstrations to further accelerate learning. Fig. 1 illustrates the core idea of IBRL, and how an IL policy is explicitly integrated in the interaction and training phase of RL. By leveraging a standalone IL policy throughout training, IBRL benefits from the IL policy's generalization and consistent support for every stage as it gradually makes progress towards solving a task. Furthermore, the modular nature of IBRL allows for a flexible framework, where we can use different architectures for each component, e.g., using different image encoders for the IL vs. RL policy, which was not possible in prior work that usually use a single architecture for both RL and IL. For instance, we show that the ResNet-18 image encoder (He et al., 2016) often yields high performance when training IL, but can lead to disastrous performance in RL, while a shallower ViT-based encoder (Dosovitskiy et al., 2020) that performs worse in IL performs much better in RL.

We evaluate IBRL on 7 sparse reward continuous control tasks: 3 increasingly difficult tasks from Robomimic (Mandlekar et al., 2021) and 4 tasks from Meta-World (Yu et al., 2019). IBRL outperforms two strong baselines, Reinforcement Learning from Prior Demonstrations (RLPD) (Ball et al., 2023) and Model-based Reinforcement Learning from Demonstrations (MoDem) (Hansen et al., 2023a). RLPD combines the idea of oversampling demonstrations with many modern sample efficient model-free RL improvements. MoDem is a model-based approach that first pretrains the policy, critic, and world model on demonstrations and data generated by the pretrained policy. Both methods have previously outperformed a range of other methods, representing the state-of-the-art among model-free and model-based methods respectively. IBRL achieves remarkable sample efficiency compared to these methods on all 7 tasks. For example, when trained from raw pixel inputs, it learns to lift a randomly positioned block with only **1** demonstration and **20K** online interaction steps. In the more complex Robomimic PickPlaceCan task, IBRL outperforms RLPD by **6.4×** under the budget of **10** demonstrations and **100K** online interactions. Videos of sample rollouts by IBRL are made available at https://sites.google.com/view/ibrl-anon.

## 2 RELATED WORK

In this section, we provide an overview of approaches that attempt to improve the sample efficiency of online reinforcement learning methods with and without access to prior demonstrations.

**Sample-Efficient RL.** A number of recent works have made significant progress in this regard from a pure RL perspective. For instance, applying regularization to the Q-function makes it possible to increase the update-to-data (UTD) ratio (i.e., the number of updates for every transition collected), which leads to faster convergence and thus higher sample efficiency. RED-Q (Chen et al., 2021) and Dropout-Q (Hiraoka et al., 2022) are two examples of methods that apply regularization to the Q-function. These techniques use multiple Q-networks and dropout in Q-networks to reduce the maximization bias in Q-function estimates (Thrun & Schwartz, 1993). However, using multiple Q-networks has been primarily tested when learning from low-dimensional states due to the high computation cost of optimizing multiple networks. When learning directly from pixel inputs, data augmentation via random shifts (Yarats et al., 2022) can instead boost performance and sample efficiency. We apply RED-Q and random shift data augmentation in IBRL to get a strong baseline, and further demonstrate that IBRL can still provide significant improvements beyond these practical strategies for improving RL sample efficiency. Additionally, in IBRL, we show that regularizing the policy network, i.e., *actor*, with dropout (Srivastava et al., 2014) also accelerates convergence.

**Model-Free RL with Prior Demonstrations.** In addition to the techniques above, a promising way to increase the sample efficiency of RL and help mitigate exploration challenges is to utilize prior demonstrations. One approach that leverages demonstrations in off-policy RL is to include the demonstrations in the replay buffer and oversample the demonstrations during training (Vecerík et al., 2017; Hester et al., 2018). Several works also add regularization to keep the learned policy similar to the prior demonstrations (Hester et al., 2018; Nair et al., 2018; Rajeswaran et al., 2018; Rudner et al., 2021; Shah & Kumar, 2021; Haldar et al., 2022). Other methods include pre-training visual representations on the demonstrations (Zhan et al., 2022) or pre-training the policy using offline RL (Hester et al., 2018; Nair et al., 2020). Recently, Ball et al. (2023) provide a systematic study of how off-policy algorithms can be modified to leverage demonstrations, and introduce Reinforcement Learning with Prior Data (RLPD), an effective approach which combines oversampling of the demonstrations with other techniques for sample-efficiency (high UTD ratio, layer normalization, Q-ensembling, and image augmentation). Given the strong performance of RLPD, we use it as a model-free baseline in this work. Unlike prior works, we train a standalone imitation policy based on the demonstrations and explicitly use it during training and inference.

**Model-Based RL with Prior Demonstrations:** Model-based RL settings typically learn a world model (often over latent representations) simultaneously with learning the policy, and use imagined rollouts from the world model to aid in policy improvement. Demonstrations can be incorporated into model-based methods as well, but simply initializing the policy via behavior cloning is not performant if the world model or critic is not also pre-trained. Therefore, Hansen et al. (2023a) introduce MoDem, a model-based RL method that uses demonstrations to pre-train a policy via behavior cloning and also pre-trains the world model and critic using demonstrations seeded from BC policy. MoDem compares favorably to model-free and model-based methods which incorporate demonstrations (Rajeswaran et al., 2018; Hafner et al., 2021; Seo et al., 2022; Zhan et al., 2022), and so we use MoDem as a model-based baseline. Like MoDem, our method trains an IL policy on a small budget of demonstrations, but the mechanism by which we use the IL policy differs in two ways. First, we keep a standalone, fixed copy of the IL policy throughout training rather than initializing the RL policy with it and fine-tuning it online. Second, our method uses the IL policy to propose actions for computing the bootstrap target as well as during online interaction, unlike MoDem which uses the IL policy to pre-train an intermediate world model. We find that our method is more computationally efficient than MoDem while also achieving superior performance.

## 3 BACKGROUND

**Reinforcement Learning.** We consider a Markov decision process (MDP) consisting of state space $s \in \mathcal{S}$, continuous action space $\mathcal{A} = [-1, 1]^d$, deterministic state transition function $\mathcal{T} : \mathcal{S} \times \mathcal{A} \to \mathcal{S}$, sparse reward function $\mathcal{R} : \mathcal{S} \times \mathcal{A} \to \{0, 1\}$ that returns 1 when the task is completed and 0 otherwise, and discount factor $\gamma$. We experiment with both RL from pixels as well as from low-

---

**Algorithm 1** Imitation Bootstrapped Reinforcement Learning (IBRL) with TD3 backbone. Major modifications w.r.t. vanilla TD3 highlighted in blue.

---

1: **Hyperparameters**: num of critics $E$, num critic updates $G$, exploration std $\sigma$, noise clip $c$
2: Train imitation policy $\mu_\psi$ on demonstrations $\mathcal{D} = \{\xi_1, \ldots, \xi_n\}$ with the selected IL algorithm.
3: Initialize policy $\pi_\theta$, target policy $\pi_{\theta'}$, and critics $Q_{\phi_i}$, target critics $Q_{\phi_i'}$ for $i = 1, 2, \ldots, E$
4: Initialize replay buffer $B$ with demonstrations $\{\xi_1, \ldots, \xi_n\}$
5: **for** $t = 1, \ldots,$ num_rl_steps **do**
6:     Observe $s_t$ from the environment
7:     Compute IL action $a_t^{\text{IL}} \sim \mu_\psi(s_t)$ and RL action $a_t^{\text{RL}} = \pi_\theta(s_t) + \epsilon, \epsilon \sim \mathcal{N}(0, \sigma^2)$
8:     Sample a set $\mathcal{K}$ of 2 indices from $\{1, 2, \ldots, E\}$
9:     Take action with higher Q-value $a_t = \arg\max_{a \in \{a^{\text{RL}}, a^{\text{IL}}\}} [\min_{i \in \mathcal{K}} Q_{\phi_i}(s_t, a)]$
10:     Store transition $(s_t, a_t, r_t, s_{t+1})$ in $B$
11:     **for** $g = 1, \ldots, G$ **do**
12:         Sample a minibatch of $N$ transitions $(s_t^{(j)}, a_t^{(j)}, r_t^{(j)}, s_{t+1}^{(j)})$ from $B$
13:         (Optional) Augment the minibatch with $M$ transitions from successful episodes
14:         Sample a set $\mathcal{K}$ of 2 indices from $\{1, 2, \ldots, E\}$
15:         For each element $j$ in the minibatch, compute target Q-value

$$y^{(j)} = r_t^{(j)} + \gamma \max_{a' \in \{a^{\text{IL}}, a^{\text{RL}}\}} \left[ \min_{i \in \mathcal{K}} Q_{\phi_i'}(s_{t+1}, a') \right]$$

$$a^{\text{IL}} \sim \mu_\psi(s_{t+1}) \text{ and } a^{\text{RL}} = \pi_{\theta'}(s_{t+1}) + \text{clip}(\epsilon, -c, c)$$

16:         Update $\phi_i$ by minimizing loss: $L(\phi_i) = \frac{1}{N} \sum_j [y^{(j)} - Q_{\phi_i}(s_t^{(j)}, a_t^{(j)})]^2$ for $i = 1, \ldots, E$
17:         Update target critics $\phi_i' \leftarrow \rho\phi_i' + (1-\rho)\phi_i$ for $i = 1, \ldots, E$
18:     **end for**
19:     Update $\theta$ with the last minibatch by *maximizing* $\frac{1}{N} \sum_j \min_{i=1,\ldots,E} Q_{\phi_i}(s_t^{(j)}, \pi_\theta(s_t^{(j)}))$
20:     Update target actor $\theta' \leftarrow \rho\theta' + (1-\rho)\theta$
21: **end for**

---

dimensional states. For simplicity, we use $s$ to generally denote the input to the policy. We use Twin-Delayed DDPG (TD3) (Fujimoto et al., 2018; Lillicrap et al., 2016) as our RL backbone.

**Imitation Learning.** We assume access to a dataset $\mathcal{D}$ of demonstrations collected by expert human operators. Each trajectory $\xi \in \mathcal{D}$ consists of a series of transitions $\{(s_0, a_0, r_0), \ldots, (s_T, a_T, r_T)\}$. The reward $r_t$ is simply 0 or 1 denoting whether a task is completed. Since the demonstrations are collected by expert humans, we can assume that at least $r_T = 1$. The goal of IL is often to train a policy $\mu_\psi$ to minimize the negative log-likelihood of data, i.e., $L(\psi) = -\mathbb{E}_{(s,a)\sim\mathcal{D}}[\log \mu_\psi(a|s)]$. In this work, we use behavior cloning (BC) with unimodal Gaussian assumption for action distribution as our IL method for its simplicity. In BC, the training objective for the policy becomes $L(\psi) = \mathbb{E}_{(s,a)\sim\mathcal{D}}[\mu_\psi(s) - a]^2$. We note that our IBRL framework can easily accommodate more powerful IL methods such as BC-RNN with Gaussian mixture model (Mandlekar et al., 2021).

## 4    IMITATION BOOTSTRAPPED REINFORCEMENT LEARNING

In this section, we first describe the IBRL algorithm. Then, we discuss two architectural improvements that can be applied independently of IBRL.

The core idea of IBRL is to first use expert demonstrations to train an IL policy and then leverage this frozen IL policy in two phases: 1) to help exploration during the online interaction phase, and 2) to help with target value bootstrapping in the Q-network training phase (as shown in Fig. 1). We focus our discussion on off-policy RL methods since they often have higher sample efficiency by effectively reusing past experience. Most popular off-policy RL methods for continuous control, such as SAC (Haarnoja et al., 2018) or TD3 (Fujimoto et al., 2018), involve training Q-networks to evaluate the action quality and training a separate policy network to generate actions with high Q-values. Samples from the policy network are used in the two phases in Fig. 1: for online interaction as well as target Q-value bootstrapping in training.

**Online Interaction:** *Actor Proposal.* In sparse reward robotics tasks, such as picking up a block, randomly initialized Q-networks and policy networks may hardly obtain any successes even after a long period of interactions, resulting in no signal for learning. IBRL helps mitigate the exploration challenge by using a standalone IL policy $\mu_\psi$ trained on human demonstrations $\mathcal{D}$. IBRL uses the IL policy to proposes an alternative action $a^{\text{IL}} \sim \mu_\psi(s)$ in addition to the action $a^{\text{RL}} \sim \pi_\theta(s)$ proposed by the RL policy at each online interaction step. Then, IBRL queries the Q-network $Q_\phi$ and selects the action with higher Q-value between the two candidates. That is, during online interaction, IBRL takes an action that provides the higher Q-value between the one proposed by the imitation policy $\mu_\psi$ and the one proposed by the RL policy $\pi_\theta$ that is being trained:

$$a^* = \underset{a \in \{a^{\text{IL}}, a^{\text{RL}}\}}{\arg\max} Q_\phi(s, a). \tag{1}$$

Here, $\psi$, $\theta$, and $\phi$ refer to the parameters of the IL policy network $\mu_\psi$, RL policy network $\pi_\theta$, and Q-network $Q_\phi$ respectively. We refer to this phase of IBRL as the *actor proposal* phase.

**RL Policy Training:** *Bootstrap Proposal.* Similarly, when training the Q-networks, we can again query the IL policy $\mu_\psi$ to propose alternative actions $a^{\text{IL}}$ for computing bootstrapping targets. The only difference is that here we use the target Q-network $Q_{\phi'}$ to evaluate the RL and IL actions. This further accelerates convergence by improving the quality of the bootstrapping targets.

$$Q_\phi(s_t, a_t) \leftarrow r_t + \gamma \underset{a' \in \{a^{\text{IL}}_{t+1}, a^{\text{RL}}_{t+1}\}}{\max} Q_{\phi'}(s_{t+1}, a') \tag{2}$$

The $a^{\text{IL}}_{t+1}$ is sampled from the policy while $a^{\text{RL}}_{t+1}$ is sampled from the target actor $\pi_{\theta'}$. $\phi'$ refers to the parameters of the target $Q$-network, which is an exponential moving average of $\phi$. We refer to this second phase of IBRL as the *bootstrap proposal* phase.

Optionally, similar to prior work (Vecerík et al., 2017), we can additionally oversample the demonstrations and successful episodes in each minibatch. Oversampling can be particularly useful if the selected IL method does not fit the data well. In this work, we use behavioral cloning (BC) as the IL algorithm and TD3 as the RL algorithm. Additionally, we use an ensemble of critic networks for the bootstrap update, following Chen et al. (2021). The detailed pseudocode of the complete IBRL method is shown in Algorithm 1. Lines 2-4 do the necessary initialization for policy, critics and replay buffer. Then lines 6-10 correspond to interacting with the environment and line 9 specifically corresponds to the *actor proposal* of IBRL. Lines 12-17 are critic updates and line 15 is the *bootstrap proposal*. Finally, lines 19-20 are policy updates, which is identical to vanilla TD3. The final output of IBRL is the hybrid policy that acts following Eq. (1).

**Benefits of IBRL.** IBRL's way of integrating IL with RL has three key advantages compared to prior model-free methods such as pretraining the policy network with human demonstrations (Haldar et al., 2022) or initializing the replay buffer with demonstrations and then oversampling them.

First, a pretrained policy network may quickly get washed out by randomly initialized critics while the IL policy in IBRL serves as an anchor that the hybrid IL+RL agent can fall back to throughout the entire training process, providing consistent support for exploration at every timestep until the RL policy finds an action that reliably outperforms the IL action in terms of the Q-value. Second, even with limited demonstrations, the IL policy may generalize surprisingly well beyond the training data with techniques such as data augmentation (Young et al., 2021) and using wrist cameras (Hsu et al., 2022). As we show later, a ResNet-18 trained on **1** demonstration using wrist camera and random shift augmentation achieves **44%** success rate in the Robomimic Lift environment. By explicitly leveraging this IL policy in both the actor proposal and bootstrap proposal phases, IBRL can benefit from these generalization capabilities. Third, the modular design of IBRL enables easily selecting the "best of both worlds" from an IL and RL standpoint. For example, we can use different network architectures that are most suited for the RL and IL tasks respectively. In Section 5.3, we show that the widely used deep ResNet-18 encoder that achieves strong performance in IL performs poorly as the visual backbone for RL, while a shallow ViT encoder that performs worse in IL works quite well in RL. IBRL's modular design also makes it straightforward to integrate some powerful IL methods such as the ones based on diffusion models (Ajay et al., 2023; Reuss et al., 2023; Chi et al., 2023) with RL, which is an exciting direction for future research.

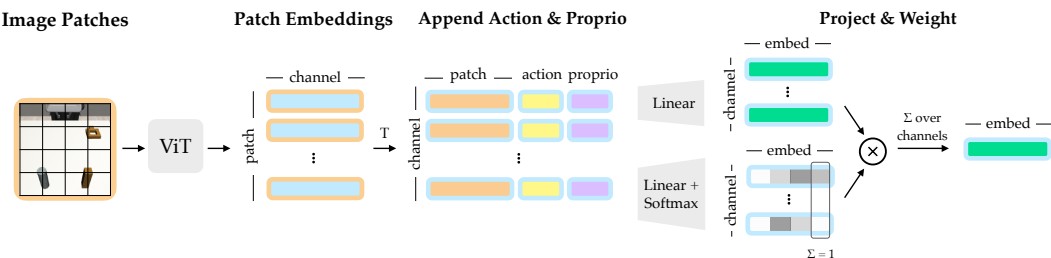

Figure 2: ViT-based Q-network with action and proprioception data appended to each feature channel.

## 4.1 KEY DESIGN CHOICES

Apart from the main innovations of IBRL, we present two architectural improvements, dropout in the actor network and a ViT-based Q-network, that further increase the performance in sparse reward control tasks considered in this paper. These techniques could also be beneficial for other methods.

**Dropout in Policy Network.** We use dropout (Srivastava et al., 2014) in the policy network (actor). Hiraoka et al. (2022) have previously applied dropout to the Q-network (critic) to reduce overfitting on the value estimate. However, to the best of our knowledge, the application of dropout in the actor has not been well-studied before. We find that adding dropout in the actor in TD3 significantly improves sample efficiency, even when other regularization techniques such as image augmentation (DrQ) (Yarats et al., 2022) or Q-ensembling (RED-Q) (Chen et al., 2021) are also present. Adding actor dropout accelerates convergence even without increasing the update-to-data (UTD) ratio (the hyperparameter $G$ in Line 11 in Algorithm 1), and requires negligible extra compute.

**ViT-Based Q-Network.** We introduce a new design for the Q-network for learning from pixels. Our network is illustrated in Fig. 2. We first use a small 3-layer ViT (Dosovitskiy et al., 2020) to convert the image to non-overlapping patch embeddings. Then we transpose the feature matrix and append actions and (optionally) proprioception data to *each channel*. A linear layer then projects the concatenated channels to larger dimensional vectors and meanwhile fuses information across patches. A separate linear projection followed by a softmax produces normalized weights along the channel dimension for the projected embeddings. The final embedding is the weighted sum of the embeddings over the channel axis, which is then fed into MLPs to compute $Q(s, a)$. Strong baselines from prior works (Yarats et al., 2022; Hansen et al., 2023b) often use a shallow ConvNet and compress its flattened output to a low dimensional feature vector of size $\mathcal{F}$ using a linear projection. Then they append the raw action to the vector before feeding it to the MLPs. This architecture usually uses small values for $\mathcal{F}$ such as $50$ or $64$ so that it does not overwhelm the action as the joint input for the subsequent MLPs. However, we observe that this narrow bottleneck limits the performance of RL, especially in complicated tasks. In comparison, our approach fuses the action with the visual observation by appending it to each channel of the visual feature early in the pipeline, eliminating the bottleneck structure and achieving better performance.

In Section 5, we perform detailed ablation for the two components discussed above. We also enable these improvements for the RLPD baseline that we compare against to emphasize the contributions from IBRL's main innovations.

## 5 EXPERIMENTS

In Section 5.1, we first discuss experimental setup and baselines, with further implementation details in Appendix F. Additionally, we plan to open-source the code. In Section 5.2, we discuss the performance of IBRL compared to baselines. Finally, in Section 5.3, we demonstrate the benefits of the key design decisions via ablations. Furthermore, we experimentally demonstrate the benefit of IBRL's modular design which enables IL and RL to use different network architectures. We also have additional ablations on the importance of the IL policy in Appendix D.

### 5.1 EXPERIMENTAL SETUP AND BASELINES

**Experimental Setup.** We evaluate IBRL on 3 tasks of increasing difficulty from the Robomimic benchmark (Mandlekar et al., 2021) and 4 tasks from Meta-World (Yu et al., 2019). All of the

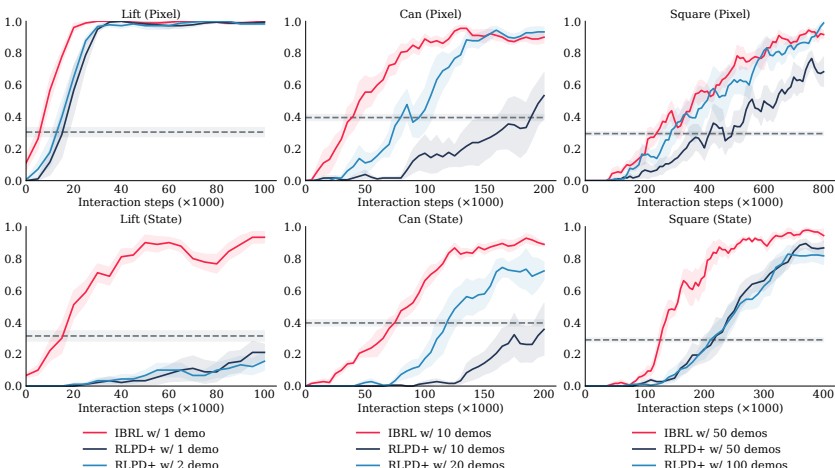

Figure 3: IBRL vs. RLPD+ using $1\times$ and $2\times$ expert demonstrations on Robomimic. We show results for both pixel- and state-based RL using *the same* pixel-based BC policy for each environment. IBRL significantly outperforms RLPD+ in all cases. It also outperforms RLPD+ when RLPD+ has twice as many demonstrations. The gray dashed lines are the average performance of BC policy used in IBRL.

experiments use sparse reward functions that return 1 when the task is completed and 0 otherwise. We experiment with RL from both pixel observations and low-dimensional states in Robomimic and from pixel observations in Meta-World. Both state- and pixel-based RL in Robomimic use *the same* pixel-based BC policy for easy comparison. The 3 Robomimic tasks are Lift, PickPlaceCan (Can) and NutAssemblySquare (Square). For the Robomimic experiments, we set $\mathcal{D}$ to consist of subsets of the proficient teleoperator demonstration datasets collected by Mandlekar et al. (2021), and use 1 demonstration for the simple Lift task, 10 demonstrations for the intermediate Can task and 50 demonstrations for the challenging Square task. The Meta-World environments are Assembly, Box Close, Coffee Push, and Stick Pull. For these environments, we set $\mathcal{D}$ to consist of 3 demonstrations generated by the scripted expert policies given by Yu et al. (2019).

**Baselines.** For Robomimic tasks, we compare against RLPD (Ball et al., 2023), which is a model-free method that oversamples the demonstrations when training the Q-function. We use our own implementation of RLPD, denoted as RLPD+, which uses the same TD3 backbone, actor dropout, and the ViT-based Q-network as IBRL to ensure a strong baseline and fair comparison. Since $\mathcal{D}$ consists of a small number of demonstrations in several of our experiments, we also dynamically expand the datasets with successful online rollouts in the RLPD+ baseline to prevent overfitting. RLPD+ uses an oversampling ratio of $0.5$, and it uses the same value for Q-ensembling size $E$ and number critic updates $G$ as IBRL. In Appendix C, we demonstrate that these modifications make RLPD+ *a significantly stronger baseline* than the vanilla RLPD. For Meta-World tasks, we additionally compare against MoDem (Hansen et al., 2023a) using the open-sourced code from the authors without modification. MoDem is a model-based method consisting of 3 phases. It first pretrains an encoder and a policy head with BC using demonstrations. Then it pretrains a latent dynamics model, a reward prediction head, and a Q-function head with the demonstrations as well as rollouts generated by the pretrained policy. Finally, it fine-tunes all components jointly using data generated by an MPC planning method (Hansen et al., 2022). We regenerate the demonstrations in two resolutions using the scripted expert policies from Yu et al. (2019) since our networks operate on a different resolution than MoDem. We reduce the number of demonstrations from 5 in original MoDem to 3 as we find that the demonstrations we generate are easier for BC to learn from. Our rerun of MoDem performs similarly or better than the original results reported in their paper despite using fewer demonstrations.

## 5.2 EXPERIMENTAL RESULTS

In this section, we detail how IBRL performs relative to the baseline methods on each environment.

**IBRL outperforms RLPD+.** Fig. 3 and Fig. 5 show the performance of IBRL compared to RLPD+ on Robomimic and Meta-World tasks respectively. This comparison highlights the algorithmic benefits of IBRL by abstracting away our design choices as both methods share the same architecture

| Environment | Lift | Can | Square |
|---|---|---|---|
| Human | 48.3 | 116.0 | 150.8 |
| BC (Pixel) | 78.0 | 134.6 | 155.8 |
| IBRL (Pixel) | 16.1 | 67.4 | 68.7 |
| Speed up IBRL vs Human | 3× | 1.7× | 2.2× |

Table 1: Mean episode length of human demonstrations and BC and IBRL rollouts trained from pixels. On average, IBRL rollouts contain 2.3× fewer steps.

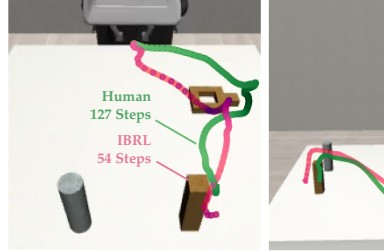

Figure 4: Samples of human demonstration and IBRL rollout for Can.

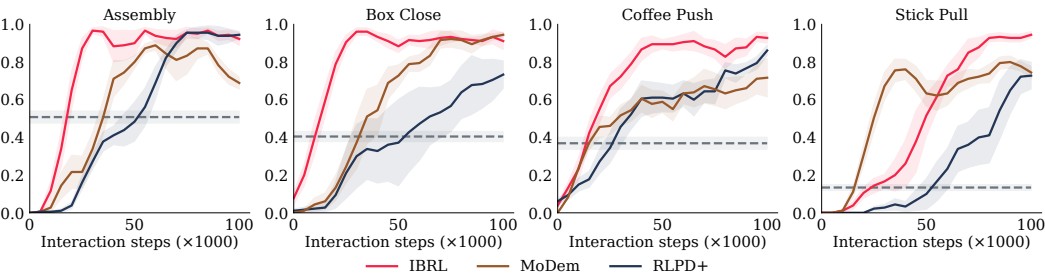

Figure 5: IBRL vs. MoDem vs RLPD+ on Meta-World. IBRL outperforms both MoDem and RLPD+ on all 4 tasks. Additionally, IBRL takes 3.2 hours to train while MoDem takes 16.7 hours on the same hardware, making IBRL **5.2× faster** in wall-clock time. The dashed lines are the average score of BC policies in IBRL.

and hyperparameters. In Robomimic tasks, IBRL significantly outperforms RLPD+ in both pixel- and state-based experiments and it even outperforms RLPD+ trained on twice as many demonstrations. In the Can task, IBRL learns to solve the task with $100K$ samples, outperforming RLPD+ by $6.4×$ with the same dataset $\mathcal{D}$ and interaction budget. Similarly, IBRL outperforms RLPD+ on all 4 Meta-World tasks. These results show that integrating an IL policy into RL using IBRL significantly improves sample efficiency compared to simply adding and oversampling demonstrations from the replay buffer.

Apart from the main takeaway that IBRL outperforms RLPD+, we also note a few interesting observations. In Lift and Can, pixel-based RL exhibits better sample efficiency for both methods. This is likely due to the combination of random-shift data augmentation and the use of wrist camera (Hsu et al., 2022), which jointly impose a strong inductive bias for pick-and-place style tasks. In Square, however, it is significantly harder to learn from pixels. Square requires both pick-and-place skills as well as precise control for aligning objects from close-up. Therefore, parts of the state are sometimes occluded, and it may require a more holistic representation from multiple camera views. Additionally, despite achieving better results than baselines, IBRL's performance on Square is also affected by the fact that BC with unimodal Gaussian does not perform well on this task (Mandlekar et al., 2021). Lastly, we demonstrate the benefit of using RL as opposed to just imitating the demonstrations, by comparing the length of trajectories rolled out from IBRL vs. human demonstrations. In Table 1, we show that the converged RL policies complete the tasks using an average of $2.3×$ fewer steps than the demonstrations, while the BC policy generally requires more steps than the human. Fig. 4 qualitatively shows the difference between efficiency of trajectories from IBRL vs. sample trajectories from a human demonstrator given the same environment initialization.

**IBRL outperforms MoDem.** Fig. 5 shows the results of IBRL and MoDem on 4 Meta-World tasks. IBRL outperforms MoDem on all 4 tasks at $100K$ interaction steps. On the Assembly, Box Close, and Coffee Push tasks, IBRL performs better than MoDem under any interaction budget. On the Stick Pull task, IBRL initially learns slower but eventually surpasses MoDem to achieve perfect success rate while MoDem converges at around $80\%$ success rate. Remarkably, IBRL solves all tasks with $100\%$ success rate while MoDem cannot reliably solve 3 out of the 4 tasks. As a model-free method without the overhead of planning and training a world-model, IBRL achieves stronger performance at a considerably smaller computation cost. Specifically, IBRL takes 3.2 hours to train while MoDem takes 16.7 hours on our hardware, making IBRL $5.2×$ faster in wall-clock time.

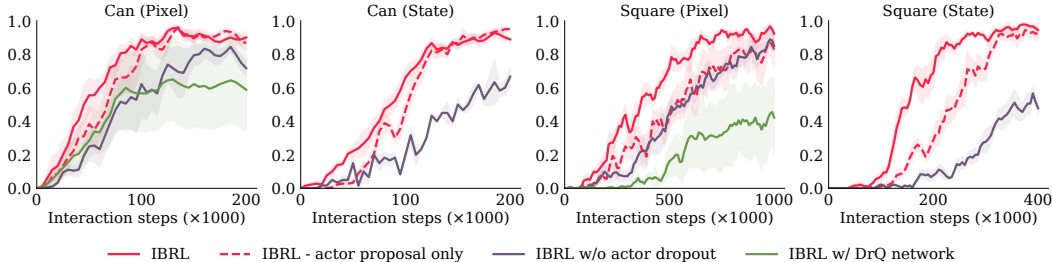

Figure 6: Ablations. *IBRL - actor proposal only* uses the IL policy only in the actor proposal phase, not in the bootstrap proposal phase. *IBRL w/o actor dropout* sets the dropout rate in the actor to 0. *IBRL w/ DrQ network* uses the architecture popularized by DrQ (Yarats et al., 2022), a strong method for pixel-based RL.

## 5.3 ABLATIONS

**Ablation of Key Components.** We perform 3 ablations to show the importance of each component in IBRL. First, we compare to a version of IBRL named *IBRL - actor proposal only* which only uses the IL policy for action proposal during online interactions and does not use IL for computing bootstrap target in training. Second, we evaluate the importance of actor dropout by comparing IBRL to *IBRL without actor dropout*. Finally, we show the benefit of our ViT-based Q-network illustrated in Fig. 2 by comparing to *IBRL with DrQ network* that uses the ConvNet-based network in DrQ (Yarats et al., 2022). The results are shown in Fig. 6. We first observe that *IBRL - actor pro-*

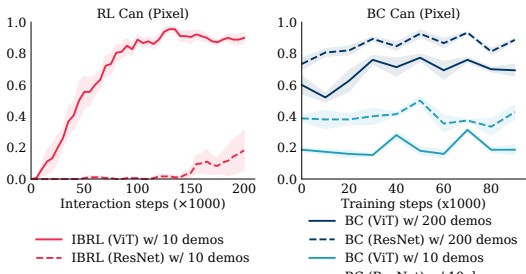

Figure 7: Performance of 3-layer ViT vs. ResNet-18 in RL and BC on Can. For BC, we show success when trained on 10 and 200 demonstrations. Although ResNet-18 is better for BC, it performs significantly worse for RL.

*posal only* performs worse than IBRL, and that the gap is larger for the harder task. This shows the importance of using the IL policy for both actor proposal and bootstrap proposal as the latter may accelerate training by producing better bootstrapping targets early on in the training process. *IBRL w/o actor dropout* performs significantly worse than IBRL, even when we use image augmentation in pixel-based RL and large Q-ensembling of 5 Q-networks in state-based RL. It is also encouraging that actor dropout improves performance without requiring more frequent gradient steps (i.e., without increasing UTD ratio), adding nearly zero computational overhead. *IBRL w/ DrQ network* also performs significantly worse than the default IBRL especially in harder tasks, which indicates that our new ViT-based design is more scalable for complicated domains.

**ViT vs. ResNet.** We empirically discover that the ResNet-18 that has been widely used in IL performs poorly in RL, as shown in Fig. 7. Note that we have replaced the BatchNorm (BN) in ResNet with GroupNorm (Wu & He, 2018), as it is a known issue that BN leads to instability in RL (Kumar et al., 2022), especially with moving average target networks. Interestingly, Fig. 7 also shows that the 3-layer ViT backbone used in RL performs worse than ResNet in BC. This suggests that RL and BC benefit from different inductive biases for the underlying networks. IBRL supports using the most suitable architectures for IL and RL thanks to its modular design.

## 6 CONCLUSION AND FUTURE WORK

We present IBRL, a novel way to use human demonstrations for sample efficient RL by first training an IL policy and using it in RL to propose actions in both online interaction and bootstrap target computation. We show that IBRL outperforms prior SoTA methods by a large margin across 7 tasks, including complex tasks such as Can and Square. While we instantiated IBRL with specific choices of IL and RL algorithms, the framework is general and can in principle be used to combine any IL method and off-policy RL method. It would be an exciting direction for future research to extend IBRL to take advantage of recent IL advancements such as diffusion policies (Reuss et al., 2023; Chi et al., 2023) or learning with hybrid actions (Belkhale et al., 2023) for even better performance.

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

# A COMPARISON WITH REGULARIZED RL FINE-TUNING ON ROBOMIMIC

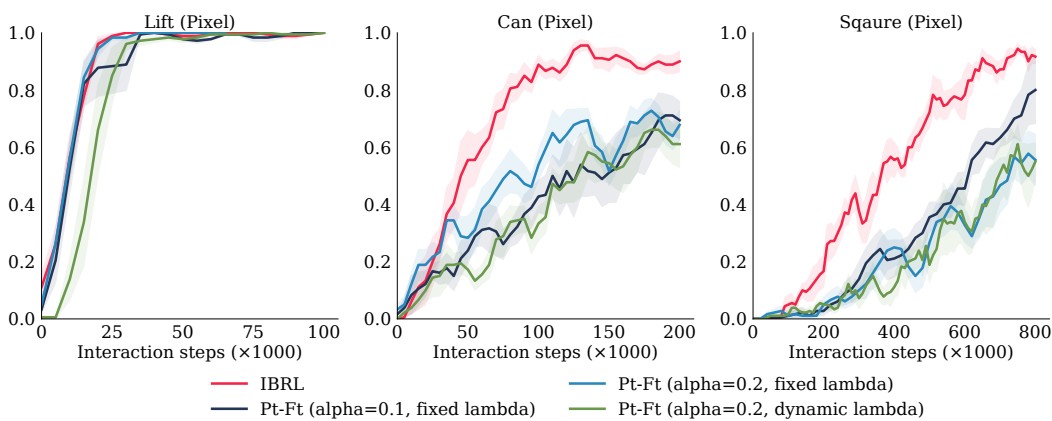

Figure 8: IBRL vs Pretraining with BC and finetuning with BC-regularized RL (Pt-Ft) on pixel-based Robomimic tasks. Here we show Pt-Ft with the best three configurations for the regularization weight $\alpha$ found through hyper-parameter tuning. IBRL outperforms all of them on the more complex Can and Square tasks without the need to tune any extra hyper-parameters.

A popular baseline in the IL+RL community is to first pretrain the policy with BC and then finetune it with RL. We refer to this method as *Pt-Ft* (pretrain-finetune). During RL finetuning, an additional BC term is added to the policy loss to prevent the catastrophic forgetting caused by bad initial critics (Nair et al., 2018). Specifically, the actor loss becomes

$$\pi = \arg\max_{\pi} \mathbb{E}_{(s,a)\sim\mathcal{D}_r} Q(s,a) - \alpha\lambda(\pi)\mathbb{E}_{(s,a)\sim\mathcal{D}_d}||a - \pi(s)||^2, \tag{3}$$

where $\mathcal{D}_r$ is the RL replay buffer, $\mathcal{D}_d$ is the demonstration dataset, $\alpha$ is hyperparameter controlling the weight of the regularization and $\lambda(\pi)$ is an optional schedule for the weight. We sweep over a range of $\alpha$s and experiment both fixed $\lambda = 1$ and dynamic $\lambda$ scheduling using soft-Q filtering from Haldar et al. (2022) where the relative strength between the pretrained policy $\pi^{BC}$ and the online RL policy $\pi^{RL}$ is used to adjust the regularization weight. Specifically, in the dynamic case

$$\lambda(\pi) = \mathbb{E}_{(s,a)\sim\mathcal{D}_r} \mathbb{1}[Q(s, \pi^{BC}(s)) > Q(s, \pi^{RL}(s))]. \tag{4}$$

Fig. 8 shows the performance of Pt-Ft in comparison with IBRL. Pt-Ft uses the same ViT-based network and actor dropout as IBRL for fair comparison. We experimented with a wide range of $\alpha$ and show the top three best performing values. In the simplest Lift task, both methods solve it with high sample efficiency. On the harder tasks, however, IBRL outperforms the best Pt-Ft configuration in terms of both sample efficiency and final performance. IBRL has two main advantages over Pt-Ft. First, it completely decouples IL from RL, allowing IL to use much powerful networks, such as the deep ResNet-18 used in this paper. As shown in Fig. 7, BC with deep ResNet outperforms BC using the ViT architecture from RL. Second, it does not require any extra tuning on the regularization, the contribution from IL and RL polices are balanced automatically by the Q-function during both online interaction and training.

# B COMPARISON WITH SQIL ON ROBOMIMIC

SQIL (Reddy et al., 2020) is a reinforcement learning with human demonstration method consisting of three parts. First, it pre-fills the replay buffer with demonstrations and label all frames with $+1$ reward. Second, it interacts with the environment and label all new interactions with $0$ reward. Third, during RL training, it samples $50\%$ of the data from the demonstrations and $50\%$ from online interactions. We implement SQIL as an additional baseline. Our implementation of SQIL uses the exact same network architecture as IBRL and uses the same TD3 backbone. Results for SQIL in comparison with IBRL and RLPD+ are shown in Fig. 9. Given the limited amount of demonstration data, SQIL does not perform well. SQIL and RLPD share the similar strategy of adding demonstrations to the replay buffer and oversampling them during training. However, SQIL performs worse than RLPD because it does not utilize the success/failure signal from the environment.

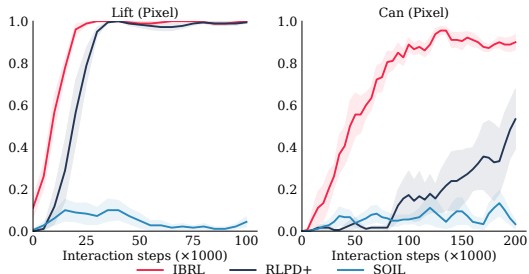

Figure 9: Performance of IBRL, RLPD+ and SQIL on Lift and Can from Robomimic. With limited amount demonstration data considered in this paper (1 for Lift and 10 for Can), SQIL fails to solve both tasks. All methods use 1 demonstration for Lift and 10 demonstrations for Can.

## C  COMPARISON BETWEEN RLPD+ AND RLPD

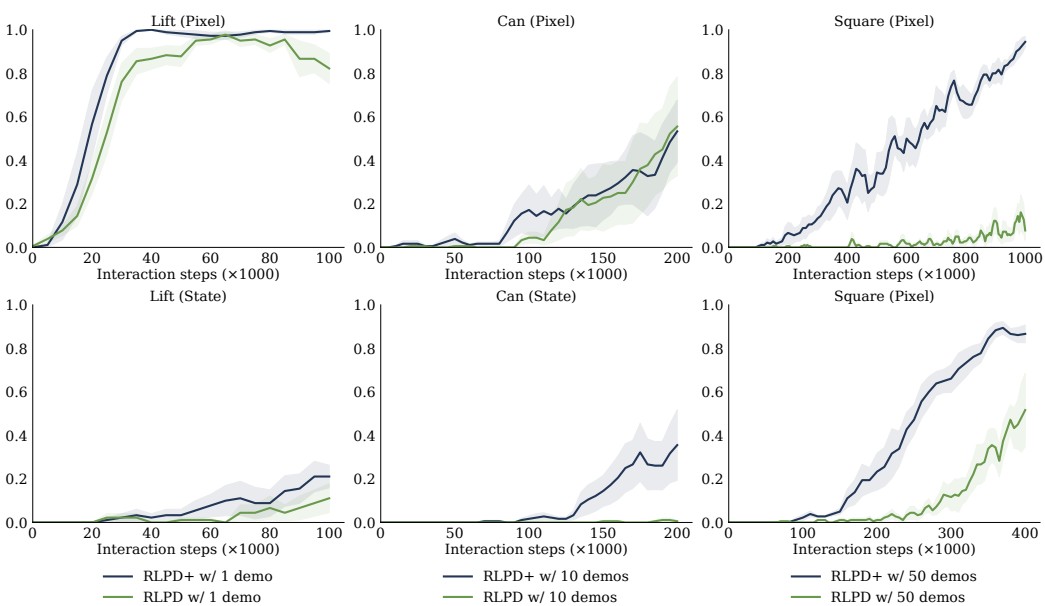

Figure 10: RLPD+ vs. RLPD on Robomimic. RLPD+ significantly outperforms RLPD in all scenarios except for Can (Pixel), where two methods perform similarly.

As mentioned in Section 5.1, our implementation of RLPD, namely RLPD+, integrates the same modifications used by IBRL. In both image- and state-based RL, RLPD+ uses actor dropout and dynamically grows the oversampling dataset by adding successful online episodes to the dataset. In image-based RL, RLPD+ uses the same ViT-based Q-network proposed in Section 4.1. We conduct experiments to show that the baseline how these modifications affect the baseline. As shown in Fig. 10, RLPD+ significantly outperforms RLPD in 5 out of the 6 scenarios evaluated and performs equally well with RLPD on the remaining one. This experiment not only further strengthens the advantages of IBRL since it outperforms a strong, properly tuned baseline, but also shows that the design choices proposed in Section 4.1 can be generally beneficial for other methods beyond IBRL.

## D  IMPORTANCE OF IL POLICY IN IBRL AT CONVERGENCE

An interesting ablation is to understand if we still need the IL policy at evaluation time and how many actions of IBRL come from the IL policy versus the RL policy. In Fig. 11, we evaluate the performance of IBRL's RL policy when used as a standalone policy. Note that both curves from each plot are from the *exact same* training runs using the full IBRL algorithm. The only difference is in

evaluation where *"IBRL RL policy only"* unrolls the RL policy while *"IBRL RL + IL policy"* unrolls both RL and IL policies and selects the better action from the ones proposed by them following the original IBRL inference rule. From the plots we can see a noticeable gap between the performance of the RL policy versus the IBRL hybrid policy that takes the better action among the ones proposed by RL and IL. The IBRL hybrid policy always performs no worse than the underlying RL policy. In Fig. 12, we show how often the IL action gets selected during evaluation as training progresses. The percentage of actions from IL generally starts from a low value, likely because the initial Q-networks are easy to exploit and so the RL policy can quickly find actions that have falsely high Q-values. Then, IBRL quickly fixes those errors of the Q-networks and starts to select many more actions from the IL policy. As the training progresses, the RL policy is able to either replicate the good actions from IL or find even better actions, leading to a steady decrease of the usage of the IL actions. From Fig. 11 and Fig. 12, it is interesting to know that the IL policy still plays an important role in IBRL even after convergence.

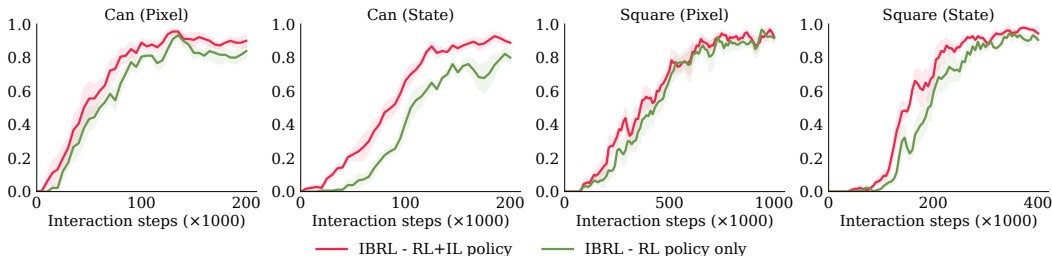

Figure 11: Performance of the RL policy in comparison with the hybrid RL + IL policy in IBRL. Note that the two curves in each plot are from the *exact same* training runs. The *IBRL - RL policy only* unrolls only the RL policy during evaluation while the *IBRL RL + IL policy* unrolls both policies and select action following the IBRL rule.

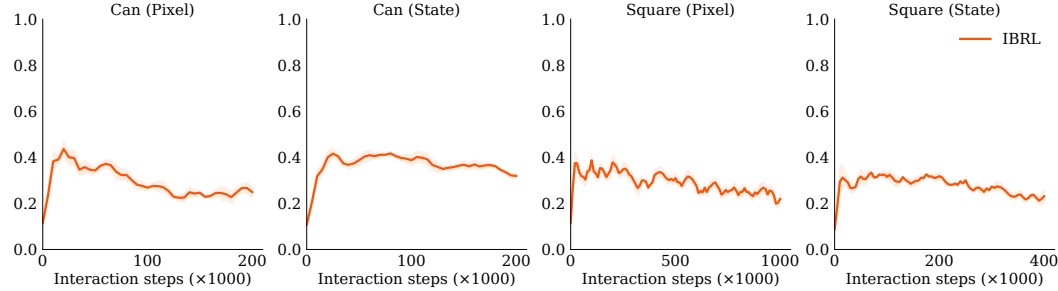

Figure 12: The percentage of actions from IL policy during evaluation of IBRL. IBRL still selects decent amount of actions ($\sim 20\%$) from the IL policy at convergence.

## E IBRL BASIC VERSUS MODEM ON META-WORLD

To understand whether the improvement of IBRL over MoDem comes from the main novelty of IBRL or the two design choices, we run *IBRL Basic*, a version of IBRL that uses a 4-layer shallow ConvNet widely used in prior works Yarats et al. (2021) and does not use actor dropout. The results are shown in Fig. 13. IBRL Basic outperforms MoDem on all 4 tasks in terms of both final performance and sample efficiency. This shows that the core idea of IBRL alone is sufficient to achieve new state-of-the-art in these tasks. IBRL Basic also performs competitively against the full version of IBRL, and it even outperforms IBRL in Stick Pull, which may seem contradictory with the ablations in Fig. 6. However, this is reasonable considering that the Meta-World tasks are simpler and have less variations than the Can and Square tasks from Robomimic so that more powerful networks and regularization are no longer necessary. A simpler network is easier to optimize, leading to faster convergence and better sample efficiency in certain tasks.

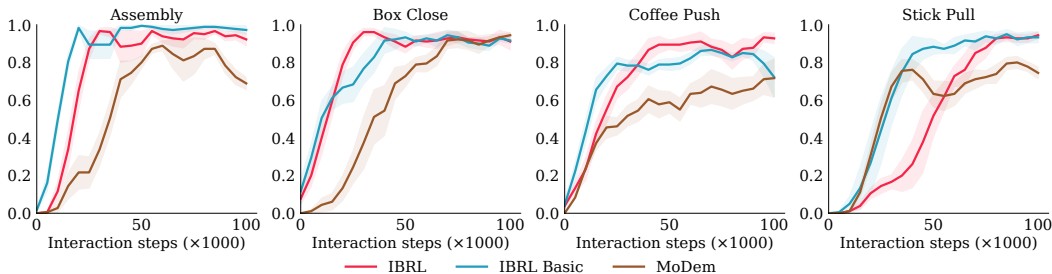

Figure 13: Performance of IBRL Basic on Meta-World. IBRL Basic uses the 4-layer ConvNet from DrQ without actor dropout. Comparing to MoDem, IBRL Basic achieves better performance with significantly less samples across *all* 4 tasks.

## F  IMPLEMENTATION DETAILS AND HYPERPARAMETERS

In this section we cover the implementation details on BC and RL training in IBRL.

The BC policies use a ResNet-18 encoder followed by MLPs. The output of the ResNet encoder is flattened and then fed into the MLPs. For all the ResNet encoders used in this paper, we replace the BatchNorm layers in ResNet with GroupNorm (Wu & He, 2018; Kumar et al., 2022) and set the number of groups equal to the number of input channels so that we can use the same network for both RL and BC since BatchNorm does not work well in RL. We train the BC policies for $100K$ steps with batch size of $256$ using Adam optimizer (Kingma & Ba, 2015) with learning rate of $1e-4$. We use random-shift data augmentation to prevent overfitting. As shown in the right panel of Fig. 7, the performance of BC policies is stable during training and we *randomly* pick a checkpoint among the top-3 checkpoints measured by evaluation score. For the Lift, Can, we use only the wrist camera image for simplicity. For the Square environment, we use the third-person camera view (agentview) as well as proprioception data because the wrist camera may not capture the goal location in this task. In all Meta-World environments, we use the same camera view as our baseline MoDem. However, unlike MoDem, we do not use proprioception data for simplicity.

For RL, we experiment with both learning from pixel observations and learning from low-dimensional state observations. However, both state- and pixel-based RL use the same pixel-based BC policy for easy comparison. For image-based RL, we follow the setup of Yarats et al. (2022) closely except that we use a target actor network, actor dropout, and the ViT-based encoder as shown in Fig. 2. The ViT encoder is shared between actor and critics while only the gradient from critic updates is used to update the encoder. We use $E = 2$ critic heads and set the critic update rate to $G = 1$. The pixel-based RL takes the same camera view as the BC policy in each environment. For state-based RL, we use Q-ensembling (RED-Q) with $E = 5$ and a higher UTD ratio $G = 5$ as we find this combination achieves good sample efficiency.

We use actor dropout with $p = 0.5$ in all environments. We set batch size to be $256$ and Adam optimizer with learning rate $1e-4$ in all experiments. For Lift, Can, and all Meta-World tasks, all $256$ samples are drawn uniformly from the replay buffer. We find that oversampling no longer improves sample efficiency under IBRL for these simple to intermediate tasks. However, for the difficult Square task, we use oversampling in IBRL similar to prior methods where $128$ samples are drawn uniformly from the replay buffer while the remaining $128$ samples are drawn from demonstrations and successful online rollouts. We do not use action repeat for Robomimic tasks. In Meta-World, we inherit the action repeat value from prior work for fair comparison. Please refer to Table 2 for a comprehensive list of hyperparameters used in our experiments.

| Parameter | Robomimic | | | Meta-World |
|---|---|---|---|---|
| | Lift | Can | Square | |
| Optimizer | | Adam | | |
| Learning Rate | | 1e−4 | | |
| Batch Size | | 256 | | |
| Discount ($\gamma$) | | 0.99 | | |
| Exploration Std. ($\sigma$) | | 0.1 | | |
| Noise Clip ($c$) | | 0.3 | | |
| EMA Update Factor ($\rho$) | | 0.99 | | |
| Q-Ensemble Size ($E$) | | 2 (pixel) / 5 (state) | | |
| Critic Update Rate ($G$) | | 1 (pixel) / 5 (state) | | |
| Actor Dropout Rate | | 0.5 | | |
| Layer Norm | | Yes | | |
| Image Size | | [96, 96] | | |
| ViT Number of Layers | | 3 | | |
| ViT Patch Size | | [8, 8] | | |
| ViT Number of Attn Heads | | 4 | | |
| ViT Patch Embed Dim | | 128 | | |
| ViT Q-network Projection Dim | | 1024 | | |
| Critic/Actor Head Hidden Dim | | 1024 | | |
| *State-based RL only* | | | | |
| Critic/Actor MLP Depth | | 3 | | N/A |
| Critic/Actor MLP Hidden Dim | 512 | 512 | 1024 | N/A |
| Number of Demonstrations | 1 | 10 | 50 | 3 |
| Oversample Success | 0 | 0 | 0.5 | 0 |
| Use Proprio | No | No | Yes | No |
| Action Repeat | | 1 | | 2 |

Table 2: Hyperparameters for IBRL.

