# OpenReview forum: "Imitation Bootstrapped Reinforcement Learning"
_ICLR.cc/2024/Conference — ICLR 2024 Conference Withdrawn Submission_

### Official Review · Reviewer_iZef · 2023-11-01

**Soundness:** 3 good
**Presentation:** 3 good
**Contribution:** 2 fair
**Rating:** 6
**Confidence:** 4

**Summary:**

The paper proposes a method named imitation bootstrapped reinforcement learning (IBRL) that uses a stand alone imitation learning model to support the learning process of a separate reinforcement learning model. The main motivation is that traditional imitation and reinforcement learning combined approach usually wants to get the best of the two methods, but often has limited benefits from the generalization of IL beyond the given demonstration data, and needs to apply additional regularizations on RL to ensure the pre-trained IL initialization does not get washed by RL fine-tune. One of the reason why these methods struggle is that the IL and RL share the same architecture, making it hard for IL to generalize to both demo and online generated data.

To mitigate these issues, the approach proposed a way to separate the training of IL and RL. In particular, the IL is trained first on the demonstration data, and then followed by two phases of RL training. The first one is the IL and RL will both propose actions during online interaction phase, and the policy select the action by either policy that has a higher Q-value. The second is that during the optimization phase of RL, the target Q-value is the max among the actions proposed by IL and RL policies. Demonstration data is also used within the RL replay buffer to accelerate learning. The authors also introduce several practical techniques to further improve the performance of IBRL, including using multiple Q-networks, dropout in Q-networks, data augmentation via random shifts, and regularizing the policy network with dropout.

This method provides the flexibility that the IL and RL model can take different architectures, and IL model also does not get confused by the noise online interaction data. The authors claimed they performed experiment to show that taking different architecture designs for IL and RL brings some benefits in terms of performance.

Experiments are performed on 7 sparse reward continuous control tasks, with increasing difficulties. Baselines such as RLPD and MoDem are used for comparisons. The author showed that the proposed IBRL method has better sample efficiency than the baseline methods on all 7 tasks. Additional ablation study finds that using IL for bootstrapping target in training phase is important for some environments and different architecture designs for IBRL and BC have huge difference in terms of performance.

**Strengths:**

1. Improved sample efficiency: IBRL is designed to learn from a limited number of expert demonstrations and achieve better performance than using BC only with the same amount of data. The authors demonstrated that IBRL outperforms existing IL+RL methods such as RLPD+ and MoDem on a range of challenging control tasks, while requiring fewer interactions with the environment.

2. Flexible network architecture selection. The algorithm design separate the training of IL and RL, and the ablation study shows that using different network designs can benefit both BC and RL, instead of forcing them the share the same network.

3. Good collection of experiments and ablation studies to support the claims. The authors performed studies to support their claim that the separate IL/RL training is beneficial, and that using IL to bootstrap the RL training target is beneficial. Ablation studies corresponding to network designs and regularization techniques also validated the design choices.

**Weaknesses:**

Some related works are not mentioned, such as "SQIL: Imitation Learning via Reinforcement Learning with Sparse Rewards" by Siddharth Reddy et al. Though this work is relatively old, but it does present a relevant approach.

The separation of IL and RL first looks novel, but the notion that different algorithms (IL vs RL) has different performances when using different architectures is a well-known thing. Therefore, it is not surprising to see the ablation study results. The regularizations used here are all from existing work, though I appreciated that the authors did experiments to show their benefits.

One confusing thing is in Figure 7, what is the point of using 10 demos to show the difference if you already have 200 demos results? I'm not sure what the authors were trying to show. On the other hand, if the cost of collecting more demos are not that high, why does it make sense to only provide one or slightly more demonstrations, which can definitely be very noisy, and then rely on RL to solve the entire problem?

The proposed IBRL method does improve on some environments, such as lift, which seems to be considered as an easier task, while the improvement on the harder task is limited. Therefore, it gives the impression that the method may overfit the easier task while its performance improvements on more complicated tasks remains unclear. On the comparison with MoDem, since the network architectures are different between IBRL and MoDem, it is hard to evaluate whether the improvement in performance comes from network design or the new method.

Since IL part is only trained on demonstration data, and is frozen at the RL training stage, it becomes unstable when the environment changes dramatically due to distributional shift. At that time, the algorithm is solely relying on RL to improve and the supervision from IL becomes less meaningful. This put this algorithm at a challenge when testing this algorithm against testing environments that are different from training tasks or testing tasks that have different distributions from training tasks. It's unclear from the paper, how different is the testing task from the training task. If it is very similar environments, there is a risk of overfitting to the environments.

**Questions:**

How does the method compare with SQIL?
How does this approach perform on tasks where testing environments are different from training environments?
How much cost is involved when collecting demonstrations? Is it more cost to collect demonstrations or is it more cost to do RL training? If we provide more demonstrations, would the method still performs better than other IL+RL methods? Since in many real world robotics applications, in general we may have some nontrivial demonstrations, and the diversity of demonstrations affect the training performance a lot. How does varying the demonstrations quantities affect the IBRL performance?

---

> ### Author Response · Authors · 2023-11-18
> **Response**
>
> Thank you for your review! We appreciate your comments that the method improves sample efficiency, that it permits flexible network architecture selection, and that it has a good collection of experiments and ablation studies to support the claims.
>
> We respond to your comments and questions below.
>
> > How does the method compare with SQIL?
>
> Thank you for the suggestion to compare to SQIL. We have implemented SQIL as an additional baseline and have included the results in **Appendix B** of the revised paper. Our implementation uses the same network architecture as IBRL and uses the same TD3 backbone. Given the limited amount of demonstration data, we find that SQIL underperforms both IBRL and RLPD+. SQIL and RLPD share a similar strategy of adding demonstrations to the replay buffer and oversampling them during training. However, SQIL performs worse than RLPD as it does not utilize the success/failure signal from the environment.
>
> > The separation of IL and RL first looks novel, but the notion that different algorithms (IL vs RL) has different performances when using different architectures is a well-known thing. Therefore, it is not surprising to see the ablation study results. The regularizations used here are all from existing work, though I appreciated that the authors did experiments to show their benefits.
>
> The primary purpose of the ablation in Figure 7 (ViT vs. ResNet on RL and IL) is to show that it is beneficial to use separate architectures in RL and IL; while this may not be surprising, it justifies the separation of the policies in IBRL. Note that prior methods such as pretraining + fine-tuning do not support the separation of RL and IL policies.
>
> Although the regularization technique (dropout) has been used in various applications, we have not found a well-documented study showing its effectiveness when applied in the *actor* network in online RL. Prior methods, such as Dropout-Q, use it only in the critic networks. We would appreciate it if the reviewer can point to specific work and we are happy to cite it.
>
> >One confusing thing is in Figure 7, what is the point of using 10 demos to show the difference if you already have 200 demos results? I'm not sure what the authors were trying to show. On the other hand, if the cost of collecting more demos are not that high, why does it make sense to only provide one or slightly more demonstrations, which can definitely be very noisy, and then rely on RL to solve the entire problem?
>
> The BC experiment in Figure 7 aims to show a gap in the BC performance for ViT and ResNet-based encoders. We run the experiment with a dataset consisting of 10 and 200 demonstrations simply to show this more conclusively—i.e., that the gap between ViT and ResNet exists regardless of whether a small or large amount of data is used for training.
>
> Robomimic was originally proposed for imitation learning and came with a large number of human demonstrations. We reduce the demonstration budget in this work because we are focused on sample efficiency of RL with few demonstrations. RL with demonstrations is a well-established area of research whose focus is on how to combine human demonstrations and autonomous RL to solve tasks with as few demonstrations and as few online interactions as possible (i.e., high sample efficiency).
>
> > The proposed IBRL method does improve on some environments, such as lift, which seems to be considered as an easier task, while the improvement on the harder task is limited. Therefore, it gives the impression that the method may overfit the easier task while its performance improvements on more complicated tasks remains unclear. On the comparison with MoDem, since the network architectures are different between IBRL and MoDem, it is hard to evaluate whether the improvement in performance comes from network design or the new method.
>
> We would argue that the improvement on the harder tasks are indeed quite significant. The hardest task we consider is the Square task in Robomimic. Consider the IBRL vs. RLPD+ curve for Square (Pixel). IBRL is significantly more sample efficient, reaching ~70% success in around 500K interactions compared to nearly 800K interactions in RLPD+. In the Can (Pixel) task, IBRL is significantly more sample efficient than RLPD+, even when RLPD+ is trained with 2x the amount of imitation data.
>
> The main places to see the benefit of IBRL are the comparison between IBRL and RLPD+ in Figure 3 and Figure 4. Because both of these methods use the same actor dropout and ViT-network. The gap there is quite large. We also show an experiment in Appendix E of IBRL-Basic vs. MoDem where neither of these methods use the dropout mechanism or ViT-network on Meta-World tasks. In both cases, our method outperforms the baselines, controlling for actor dropout and architecture.

---

> > ### Author Response · Authors · 2023-11-18
> > **Response cont.**
> >
> > > Since IL part is only trained on demonstration data, and is frozen at the RL training stage, it becomes unstable when the environment changes dramatically due to distributional shift. At that time, the algorithm is solely relying on RL to improve and the supervision from IL becomes less meaningful. This put this algorithm at a challenge when testing this algorithm against testing environments that are different from training tasks or testing tasks that have different distributions from training tasks. It's unclear from the paper, how different is the testing task from the training task. If it is very similar environments, there is a risk of overfitting to the environments.
> >
> > > How does this approach perform on tasks where testing environments are different from training environments?
> >
> > We note that we are not making any additional assumptions about the train/test distribution than prior work. We also use standard benchmarks for our evaluations (Robomimic and Metaworld) which exhibit some randomness in the scene configuration, but test environments are sampled from the same distribution as train environments.
> >
> > As you note, IL is trained on only a small amount of demonstration data and then frozen. Since the observations during RL can differ from the small number of observations seen during IL, we performed an experiment to examine how often the IL policy’s proposals are used during evaluation of IBRL. IBRL still learns to utilize actions from IL, as we shown in Figure 12 in Appendix D.
> >
> > When the test environment is significantly different from the training environment, most learning based methods would suffer, and the utility of demonstrations in the training environment would decrease for all methods. It is not in the scope of our work to address significant distribution shifts.
> >
> > > How much cost is involved when collecting demonstrations? Is it more cost to collect demonstrations or is it more cost to do RL training?
> >
> > Depending on the difficulty of the task, collecting hundreds of demonstrations can require many hours of a human demonstrator’s time. Human demonstrators also need time for training and practicing to provide high quality data. It is unclear how to compare the cost of collecting demonstrations and the cost of doing RL training since they require different resources. Given that the main downside of IL methods is the costliness of collecting demonstrations, and the main downside of RL methods is sample inefficiency, we believe that using a small number of prior demonstrations to increase the sample-efficiency of RL strikes a good balance of these costs.
> >
> > > If we provide more demonstrations, would the method still perform better than other IL+RL methods?
> >
> > We would expect that the gap between different methods will generally shrink as more demonstrations are provided because the BC's performance on more data, which can be roughly seen as the the lower bound of these IL+RL method, will definitely improve with more data. However, the IBRL framework can benefit from both _more data_ as well as _better training techniques_ for IL, which is better than methods that only benefit from more demonstrations, such as RLPD. We also point to the results in Figure 3, which shows that IBRL performs as well or better than RLPD trained with _twice_ as many demonstrations as IBRL.
> >
> > > Since in many real world robotics applications, in general we may have some nontrivial demonstrations, and the diversity of demonstrations affect the training performance a lot. How does varying the demonstrations quantities affect the IBRL performance?
> >
> > Unlike prior methods that directly use the demonstration data in the replay buffer or use the demonstrations for regularization during RL, IBRL does not directly depend on demonstration quality and diversity. Instead, it depends on how well the IL method can handle data with different qualities. Recent advancements in learning from demonstrations, such as diffusion policy and offline RL, are able to handle diverse data much more effectively than simple BC, and these IL or offline RL advancements can be more easily integrated into IBRL. Therefore, compared to prior methods, we expect that IBRL’s approach of using a separate IL policy is better equipped to handle data quality degradation.

---

> > > ### Author Response · Authors · 2023-11-22
> > > **Comment**
> > >
> > > Thank you again for the thoughtful review. Please let us know if there is anything else we can clarify before the rebuttal deadline.

---

> > > > ### Comment · Reviewer_iZef · 2023-11-23
> > > > **thanks for the rebuttal!**
> > > >
> > > > I read the authors' rebuttal, as well as other reviewers' comments, and it looks to me that this method empirically makes sense but in some cases where the IL performs bad, it heavily relies on RL to improve, so there seems to be not enough or significant gain from using this approach. So I will keep my rating.

---

### Official Review · Reviewer_8Ymy · 2023-11-01

**Soundness:** 2 fair
**Presentation:** 2 fair
**Contribution:** 2 fair
**Rating:** 5
**Confidence:** 3

**Summary:**

In this work, the authors present a reinforcement learning algorithm, Imitation Bootstrapped Reinforcement Learning (IBRL), which uses an imitation learning policy to provide alternative actions for exploration in the environment and for value function iteration. Empirically, IBRL outperforms competitive benchmarks in challenging continuous control tasks.

**Strengths:**

1.The idea of maintaining a frozen imitation learning agent prevents the knowledge obtained from expert demonstrations from being washed out, and this approach has proven successful in practice.

2.The dropout policies and ViT-Based Q-Network, accompanied by ablation studies, provide useful observations for practical implementations.

**Weaknesses:**

The lack of theoretical analysis diminishes the paper's contribution. It would be beneficial to decompose the reinforcement learning problem into imitation learning and reinforcement learning components, aiming for a 'best of both worlds' guarantee, as seen in reference [1].

Using different policies for imitation and reinforcement learning might not be sufficiently motivating. This introduces extra overhead and complicates hyperparameter tuning.

The fixed imitation learning policy doesn't leverage additional observations from reinforcement learning, where more robust frameworks, like generative adversarial imitation learning, could be naturally incorporated.

The bootstrap proposal for RL is not novel and might not be the best choice to emphasize the innovation.

[1] Cheng C A, Yan X, Wagener N, et al. Fast policy learning through imitation and reinforcement[J]. arXiv preprint arXiv:1805.10413, 2018.

**Questions:**

Same as listed in the weakness above. Additionally, please provide full result plots for all experiments in the appendix to further justify the findings.

---

> ### Author Response · Authors · 2023-11-18
> **Response**
>
> Thank you for your review! We agree with your assessment about the importance of maintaining a frozen imitation learning agent, and the benefit of using actor dropout and a ViT-based Q-network.
>
> We address your comments below.
>
> > Please provide full result plots for all experiments in the appendix to further justify the findings
>
> We have significantly expanded the Appendix to include additional experiments. These include:
> * Appendix A: A new baseline that pretrains the encoder and actor head using BC, and then fine-tunes using RL with a BC regularization loss
> * Appendix C: Comparison between RLPD+ and vanilla RLPD to show that the proposed design choices also strengthen the baseline.
> * Appendix D: New empirical analysis on the important of IL and how often the IL proposals are used during RL training.
> * Appendix E: New comparison of IBRL-Basic vs. MoDem to show that the core idea of IBRL alone is sufficient to outperform this model-based method is more computationally expensive.
>
> >Using different policies for imitation and reinforcement learning might not be sufficiently motivating. This introduces extra overhead and complicates hyperparameter tuning.
>
> A strong benefit of allowing different policies for IL and RL is that each method may benefit from different inductive biases in the network architectures. For example, as we show in Figure 7, a ResNet-18-based encoder can vastly underperform a ViT-based encoder on the Can task in Robomimic.
>
> We respectfully disagree that IBRL significantly complicates hyperparameter tuning. Many work in RL with demonstrations involving training with both IL and RL, such as pretraining with IL and finetuning with RL, as well as the LOKI method mentioned by the reviewer. On top of IL training, IBRL requires no extra hyperparameters to to balance IL and RL. In comparison, other approaches, such as  regularizing RL policies to be close to a set of demonstrations (in Appendix A),  can be sensitive to hyperparameters.
>
> >The fixed imitation learning policy doesn't leverage additional observations from reinforcement learning, where more robust frameworks, like generative adversarial imitation learning, could be naturally incorporated.
>
> It is true that the imitation learning policy is fixed and does not leverage additional observations from reinforcement learning, but we do not believe this is necessarily a problem. Keeping the imitation learning policy fixed allows us to maintain access to knowledge learned from the prior demonstrations, while avoiding the problem where the pre-trained policy could get washed out by online interactions. Further, keeping the imitation policy fixed reduces the number of moving parts in the algorithm, and does not introduce additional hyperparameters. In contrast, methods like GAIL can be challenging to apply in complex domains without significant tuning.
>
> >The bootstrap proposal for RL is not novel and might not be the best choice to emphasize the innovation.
>
> If you have a specific reference in mind, we would appreciate it if you could let us know; we are happy to explain how our work differs.
>
> >The lack of theoretical analysis diminishes the paper's contribution. It would be beneficial to decompose the reinforcement learning problem into imitation learning and reinforcement learning components, aiming for a 'best of both worlds' guarantee, as seen in reference [1].
>
> Our focus in this work is to propose an algorithm for utilizing prior demonstrations in RL with strong empirical performance across a variety of environments. While we acknowledge the lack of theoretical analysis, we note that a variety of prior works in this area (including the baselines we compare to) have a similar goal of demonstrating empirical performance across different domains. Our method is simple to implement and outperforms baselines in terms of both sample efficiency and converged performance. Given this, we believe the algorithm is a significant empirical contribution, and would be beneficial for the ICLR community.

---

> > ### Author Response · Authors · 2023-11-22
> > **Comment**
> >
> > Thank you again for the thoughtful review. Please let us know if there is anything else we can clarify before the rebuttal deadline.

---

> > ### Comment · Reviewer_8Ymy · 2023-11-23
> > **Thanks for your explanation.**
> >
> > Dear authors,
> >
> > I agree that my initial assessment of the contribution part may have been overly critical. Consequently, I have increased my score to 5.
> >
> > Regarding the imitation learning parts, I concur that integrating a frozen imitation learning policy does not introduce significant overhead, whereas using adversarial imitation learning could lead to other complications.
> >
> > Concerning the bootstrapping proposal, I believe the bootstrap occurs at line 8 of Algorithm 1, which seems to be included in previous work [1] (Randomized Ensemble Double Q-learning, line 7), as you mentioned in your paper: "Additionally, we use an ensemble of critic networks for the bootstrap update, following Chen et al. (2021)." Would you agree with this observation?
> >
> > As a final remark, I would like to point out another related work that also combines offline imitation learning and reinforcement learning [2]. It might be worthwhile to mention this in the related works section.
> >
> > Best, 8Ymy
> >
> > [1] Chen X, Wang C, Zhou Z, et al. Randomized ensembled double q-learning: Learning fast without a model[J]. arXiv preprint arXiv:2101.05982, 2021.
> > [2] Lu Y, Fu J, Tucker G, et al. Imitation is not enough: Robustifying imitation with reinforcement learning for challenging driving scenarios[J]. arXiv preprint arXiv:2212.11419, 2022.

---

> ### Author Response · Authors · 2023-11-23
> **Response**
>
> Thank you for considering our response and additional experiments, and for increasing your score!
>
> Regarding your query:
>
> > Concerning the bootstrapping proposal, I believe the bootstrap occurs at line 8 of Algorithm 1, which seems to be included in previous work [1]
>
> The “bootstrap proposal” in our work is in **Line 15** of Algorithm 1 (corresponding to the rightmost plot titled “bootstrap proposal” in Figure 1), where the “bootstrap target” for the Q-function is computed by taking actions from both the RL policy and IL policy into account. **Line 8** of Algorithm 1 corresponds to sampling a subset of _K_ critics from the total number of _N_ critics, but we do not claim “critic ensemble” as our contribution. **Line 9** of Algorithm 1 corresponds to the “actor proposal” (corresponding to the middle plot titled “actor proposal” in Figure 1).
>
> In the rebuttal revision PDF, our novel contribution is highlighted in blue, mainly line 2, 7, 9, 15. We believe that Line 15 is quite different from prior work. In ablations (Figure 6, red solid lines vs red dashed lines) we have also shown that Line 15 further improves sample efficiency, especially for harder tasks. We would appreciate it if you could reevaluate our novelty based on these points.
>
> Thank you for the new related work. We will add it in the related work section in the next revision.

---

### Official Review · Reviewer_k6EZ · 2023-11-02

**Soundness:** 2 fair
**Presentation:** 3 good
**Contribution:** 2 fair
**Rating:** 6
**Confidence:** 4

**Summary:**

This paper introduces a sample efficient reinforcement learning framework where an imitation learning policy trained on demonstrations proposes an alternative action during online exploration and target value bootstrapping. The key difference from past work on this literature is the use of a standalone IL policy throughout the entire training process. The modular design of the proposed framework also enables different network architectures for IL and RL components. In addition, dropout in policy networks and ViT-based Q-network are technical improvements that enhance performance in sparse reward control tasks. Experiment results show that this framework outperforms previous baselines in 7 robot control tasks in simulation and that the flexible architecture benefits IL and RL policies by choosing different encoders, respectively.

**Strengths:**

- The paper is well organized and written.
- The authors provide a reasonable and flexible algorithm that integrates IL and RL policies while preserving their advantages. To my knowledge, this framework seems new and would be interesting to the community.
- The performance of this algorithm is promising in the experimental tasks chosen in the paper. Ablation results demonstrate the importance of each component within the framework.

**Weaknesses:**

My primary concern and questions lie in the experiments.
- This paper investigates 7 robot control tasks and employs two test environments for two baseline methods. However, the reasons for this setup are not clearly provided. I am curious about why these tasks are selected, why comparisons are not made with the two baselines on all tasks, and how IBRL performs on other various robot control tasks.
- One of the baselines used is RLPD+, which is an improved implementation of the RLPD algorithm by the authors. A further comparison with the original RLPD would make the experimental results more persuasive and demonstrate the effectiveness of the two techniques in RLPD.
- The authors provide videos of sample rollouts of one task, showing that trained IBRL can finish the task faster than humans. Presenting more demos would give a more intuitive understanding of the difficulty of these tasks and the performance of the proposed framework.
- One benefit of the proposed approach is that "the IL policy may generalize surprisingly well beyond the training data." Although an explanation is provided, I am still confused about the supporting experimental results and how the result is connected to the claim.

**Questions:**

Please refer to the questions raised in the Weakness part.

---

> ### Author Response · Authors · 2023-11-18
> **Response**
>
> Thank you for your review! We appreciate your comments that the paper is well organized and written, that the algorithm is reasonable, flexible, new, and interesting, and that the performance of the algorithm is promising.
>
> We respond to your points below.
>
> > This paper investigates 7 robot control tasks and employs two test environments for two baseline methods. However, the reasons for this setup are not clearly provided. I am curious about why these tasks are selected, why comparisons are not made with the two baselines on all tasks, and how IBRL performs on other various robot control tasks.
>
> We have added new results showing the performance of RLPD+ on the 4 Meta-World tasks so that we can compare all methods on this benchmark. Additionally, we also implemented a new baseline where we first pretrain the policy with BC and then fine-tune it with RL. During fine-tuning, we regularize the policy with an additional BC loss. Please see the detailed description and results in Appendix A. We swept over a range of hyperparameters for the regularization weight and IBRL outperforms the best of them. In addition, IBRL does not need to adjust the regularization weight at all.
>
> In this paper, we focus on control tasks with sparse reward so we do not run experiments on locomotion tasks with dense reward such as DMControl. Robomimic is a good benchmark containing human-operated demonstrations. Lift, Can and Square represent three levels of difficulties so they are a good test combination to show the performance of IBRL under different difficulties. Square is sufficiently hard for all the RL methods we have considered in this paper, requiring 1M interactions to converge. We additionally perform experiments on Meta-World tasks so that we can compare with prior works (MoDem) using their original, well-tuned implementation. We randomly picked 4 tasks due to computational constraints.
>
> We do not run MoDem on Robomimic because
> 1) It underperforms IBRL on a simpler benchmark (Meta-World).
> 2) It is more complex and contains lots of hyperparameters to tune.
> 3) It is significantly more expensive to run. Given that it takes 16.7 hours on Meta-World, we estimate that it will take 26 hours per run on Can (Pixel) and 130 hours per run on Square (Pixel).
>
> > One of the baselines used is RLPD+, which is an improved implementation of the RLPD algorithm by the authors. A further comparison with the original RLPD would make the experimental results more persuasive and demonstrate the effectiveness of the two techniques in RLPD.
>
> Thank you for the suggestion. We have added a new plot (Figure 10) in the Appendix C to compare RLPD+ against vanilla RLPD. We note that the improvements brought by our two design choices significantly strengthen the vanilla RLPD baseline, especially in harder tasks such as Square.
>
> >The authors provide videos of sample rollouts of one task, showing that trained IBRL can finish the task faster than humans. Presenting more demos would give a more intuitive understanding of the difficulty of these tasks and the performance of the proposed framework.
>
> Thank you for the suggestion. We will update the website accordingly.
>
> >One benefit of the proposed approach is that "the IL policy may generalize surprisingly well beyond the training data." Although an explanation is provided, I am still confused about the supporting experimental results and how the result is connected to the claim.
>
> In the revised paper (Figure 3), we have added a horizontal dashed line to each plot showing the performance of the IL policy. If we take a closer look at results for the Lift task, the IL policy’s performance is quite strong given that it only uses one episode for supervision, thanks to the combination of using an in-hand camera and random-shift data augmentation. Note that Pixel-based RL uses the same camera and augmentation but state-based RL does not.
>
> On the contrary, an IL policy trained on states may not generalize as well. For example, if we train IL policy on state using the same 1 episode of demonstration, the IL policy achieves 0 score in evaluation. This can also explain why the gap between IBRL and RLPD+ in Lift (State) is significantly bigger than that in Lift (Pixel) because in both cases, IBRL uses the same pixel-based IL policy that generalizes well beyond the 1 episode that it is trained on. This experiment shows that if IL policies generalize beyond their training data, IBRL can benefit more from it than just putting those data into the replay buffer. As IL methods continue to improve, IBRL is better positioned to take advantage of those improvements.

---

> > ### Author Response · Authors · 2023-11-22
> > **Comment**
> >
> > Thank you again for the thoughtful review. Please let us know if there is anything else we can clarify before the rebuttal deadline.

---

> > > ### Comment · Reviewer_k6EZ · 2023-11-22
> > > **Thanks for the responses.**
> > >
> > > I thank the authors for the explanation and additional experiments, which addressed most of my previous concerns. I maintain a positive rating of this work.

---

### Official Review · Reviewer_GwQ8 · 2023-11-09

**Soundness:** 3 good
**Presentation:** 3 good
**Contribution:** 2 fair
**Rating:** 5
**Confidence:** 3

**Summary:**

The paper introduces Imitation Bootstrapped Reinforcement Learning (IBRL), an innovative approach that combines Imitation Learning (IL) with Reinforcement Learning (RL) to enhance sample efficiency and performance in RL tasks. IBRL utilizes expert demonstrations to train an IL policy, which then informs the action selection and target value bootstrapping in RL. The framework demonstrates good results on several continuous control tasks, significantly outperforming the selected baselines.

**Strengths:**

1. The paper is well-organized and written with a clear structure, making it accessible to readers with a background in machine learning and RL. The explanations of the IBRL framework, the role of expert demonstrations, and the architecture of the Q-network are coherent and logical. Figures and algorithmic descriptions aid in understanding the proposed method.
2. The significance of this work lies in its potential impact on the field of RL, particularly in domains where sample efficiency is critical, such as robotics or any environment where interaction is costly or risky.

**Weaknesses:**

1. While the paper demonstrates the effectiveness of IBRL, it will benefit from a broader range of baselines, including recent advancements in both IL and RL. At least the baseline algorithms that pretrain policies via behavior cloning should be implemented for the Robomimic benchmark.   Additionally, similar enhancements as applied in RLPD+ should be adopted for the MoDem benchmark to ensure consistency and a fair comparison.

2. The paper would benefit from a more detailed theoretical analysis that elucidates the necessity and efficacy of the proposed method.  The solution indeed generally makes sense but as an academic paper, we need more analysis or religious modeling to show its advantage from the methodology perspective. Experiments in Figure 6 also seem to indicate that the main benefit comes from the dropout mechanism and DrQ network instead of the actor selection and bootstrapping pipeline.

**Questions:**

What if the Q value is underestimated under some actions? In the current pipeline, it seems that these actions can never be selected and thus the better action can never be learned by RL, since the max-Q mechanism will filter these actions and replace them with the actions in IL policy.

---

> ### Author Response · Authors · 2023-11-18
> **Response**
>
> Thank you for your review! We appreciate your comments that the paper is “well-organized and written with a clear structure”, and that “the explanations of the IBRL framework, the role of expert demonstrations, and the architecture of the Q-network are coherent and logical.”
>
> We include responses to your points below:
>
> >At least the baseline algorithms that pretrain policies via behavior cloning should be implemented for the Robomimic benchmark.
>
> Thank you for the suggestion! We have included a new baseline in the revised PDF. Please see **Appendix A** for details.
>
> In short, we first pretrain the policy (encoder and actor head) with BC, and then run RL with the pretrained initialization as well as an additional BC regularization. We experimented with various values on the weight of the BC regularization, as well as dynamic annealing the weight with soft-Q filtering. IBRL outperforms the best of these.
>
> Additionally, we would like to emphasize that IBRL requires no tuning to balance the RL and IL components, making it highly desirable for real world applications where tuning is expensive.
>
> >Additionally, similar enhancements as applied in RLPD+ should be adopted for the MoDem benchmark to ensure consistency and a fair comparison.
>
> The two main enhancements we made in RLPD+ to strengthen the RLPD baseline were a change in network architecture and the addition of actor dropout. We do not apply those enhancements for MoDem because it is a quite different algorithm involving a world model with latent dynamics; it is not straightforward to control the network architecture to be exactly the same. As for actor dropout, the actor in MoDem plays a less important role since the actions are selected using TD-MPC instead of sampling from the actor.
>
> However, we do agree that controlling for enhancements such as architecture and regularization when comparing to MoDem is a valid suggestion. Therefore, we have included a new experiment in the **Appendix E** in which we compare IBRL-Basic against MoDem. IBRL-Basic uses the small 4-layer ConvNet from DrQ and does not use actor dropout. As shown in Figure 13, IBRL-Basic clearly outperforms MoDem on all 4 tasks without the modifications. The actor dropout and ViT-based critics are mainly proposed to solve harder tasks such as Can and Square in Robomimic. Given the simplicity of Meta-World tasks, IBRL can achieve new SOTA performance without them. This new experiment strengthens the results of the core IBRL method.
>
> >The paper would benefit from a more detailed theoretical analysis ...
>
> Our focus in this work is to propose an algorithm for utilizing prior demonstrations in RL with strong empirical performance across a variety of environments. While we acknowledge the lack of theoretical analysis, we note that a variety of prior works in this area (including the baselines we compare to) have a similar goal of demonstrating empirical performance and sample-efficiency across different domains. Our method is simple to implement and outperforms baselines in terms of both converged performance and sample efficiency. Given this, we believe the algorithm is a significant empirical contribution, and would be beneficial for the ICLR community.
>
> >Experiments in Figure 6 also seem to indicate that the main benefit comes from the dropout mechanism and DrQ network instead of the actor selection and bootstrapping pipeline.
>
> We would like to clarify a few important points.
>
> In Figure 6, we focus on the harder tasks from Robomimic, Can and Square. In these tasks, the capacity of the network is important, which is why switching from DrQ to ViT is necessary to unlock further benefits of the core IBRL method.
>
> The main plot that shows the benefits of IBRL are Figure 3 and Figure 4 (IBRL vs. RLPD+) where both of these methods use the same dropout mechanism and ViT-network (on Robomimic tasks, which are the harder set of tasks). In the newly added Appendix E, we have IBRL-Basic vs. MoDem where neither of these methods use the dropout mechanism or ViT-network (on Meta-World tasks, which are the easier set of tasks). In both cases, our method outperforms the baselines, controlling for dropout mechanism and architecture.
>
> Together, these experiments show both that (a) our design choices described in Section 4.1 are helpful for performance especially on harder tasks and (b) IBRL outperforms baselines when controlling for these design choices.

---

> > ### Author Response · Authors · 2023-11-18
> > **Response cont.**
> >
> > >What if the Q value is underestimated under some actions? In the current pipeline, it seems that these actions can never be selected and thus the better action can never be learned by RL, since the max-Q mechanism will filter these actions and replace them with the actions in IL policy.
> >
> > Even if the Q-value is underestimated under some actions, and the max-Q mechanism replaces these actions with proposals from the IL policy, it is not the case that better actions can never be learned by RL. In DDPG, TD3, SAC, and related algorithms, the RL actor is trained directly to choose actions with high Q-values regardless of how actions are selected online (See line 19 in Algo.1). Similarly, the Q-network is trained to compute target values of actions sampled from the replay buffer, regardless of how they are selected online. Essentially, there is no "stop gradient" effect when IL action is chosen during online interaction.

---

> ### Comment · Reviewer_GwQ8 · 2023-11-21
>
> The authors say the RL actor is trained directly to choose actions with high Q-values regardless of how actions are selected online. In fact, RL actor is trained to choose actions that are **estimated to have a higher Q-value**. If the optimal action's Q-value is underestimated, then it will never be selected, as the hard arg-max-Q strategy in  algorithm (line 9) will always replace these actions with the IL policy's action. In standard RL algorithm, we do not have such problem since the exploration policy will also select the not-optimal actions in the process of data interaction.
>
> Based on the response, I have another question: when evaluating the algorithms, are the baselines' actions selected under the stochastic strategy or the deterministic strategy?

---

> > ### Author Response · Authors · 2023-11-21
> > **Response**
> >
> > Thanks for the thoughtful comment. In a tabular case with a finite number of actions per state, the underestimated Q-values is an issue. In continuous control with deep neural networks, this issue may not be a major problem because:
> > 1. With infinite action space and sparse reward, the main challenge in practice is that RL cannot find any useful actions, let alone finding optimal actions. IBRL makes it easy to find actions that are at least as good as the BC actions, which helps exploration and thus improves sample efficiency.
> > 2. If there exist better actions with higher Q estimates, IBRL can find them and improve. It may not find _the optimal_ action at all timesteps due to the bias imposed by the argmax operation, but it is sufficient to solve the tasks that prior methods cannot solve under the same number of demonstrations.
> > 3. Given the infinite action space, the stochasticity in neural network updates and the fact that updating neural networks for certain input-output pairs also changes the output for other correlated inputs, it is unlikely for all better actions to be underestimated simultaneously. If the training does get stuck, an easy fix is to add exploration on top of the output action of line 9.
> > 4. Most existing IL + RL methods, including implicit ones such as RLPD, impose certain degrees of biases to the RL in exchange for better exploration and sample efficiency. Hand engineered dense reward (reward shaping) for vanilla RL also imposes human priors to the learning. These biases affect the optimality of the final policy at convergence. Among them, IBRL solves all tasks (achieves near 100% success rate) with significantly less samples while other methods cannot solve all tasks reliably.
> >
> > During evaluation, IBRL and all baselines select actions greedily, i.e. deterministically.
> >
> > Please also let us know if the new baselines and experiments have addressed other concerns raised in the original review.

---

### Author Response · Authors · 2023-11-18
**Summary of Additional Experiments During Rebuttal Period**

We thank all reviewers for their valuable feedback. It greatly helps us to improve this paper. We have conducted several new experiments to address the reviewers' concerns. Most new experiments are added to the appendix with minimal change to the main paper so that it is easier to pick up the most relevant information.

Specifically, the new experiments are:

**Updated Figure 5** [suggested by k6EZ]: We include RLPD+ results on Meta-Worlds tasks to have a common domain that includes all baselines.

**New Appendix A** [suggested by Reviewer GwQ8]: We compare with a new baseline where we first pretrain the RL policy with BC and then finetune it with RL. During finetuning, we add additional BC regularization loss to avoid catastrophic forgetting. We have tuned this baseline with different regularization weights and IBRL outperforms the best among them without the need of tuning any hyperparameter that balances BC and RL.

**New Appendix B** [suggested by Reviewer  iZef]: We compare with SQIL. Unfortunately SQIL fails to solve the tasks given the limited number of demonstrations considered in this paper.

**New Appendix C** [suggested by Reviewer k6EZ]: We compare RLPD+ against vanilla RLPD. RLPD+ outperforms RLPD on the Robomimic tasks. The gap between RLPD+ and RLPD is especially large in the hardest Square task, which echoes the observations from IBRL ablations. This shows that the two design choices can be generally helpful for algorithms beyond IBRL.

**New Appendix D**: We show that IL policy still plays an important role in IBRL at convergence by showing that the hybrid IL+RL policy outperforms the RL policy alone. We demonstrate that actions from the IL policy are frequently used throughout the training, even though the data distribution during online exploration could be quite different from the distribution that IL is originally trained on.

**New Appendix E** [suggested by GwQ8]: We compare IBRL Basic, the IBRL variant without the two new design choices, against MoDem for a fair comparison. IBRL Basic can still outperform MoDem significantly on these tasks.

Please see individual comments where we address each reviewer’s comments in detail.

---

### Comment · Area_Chair_KoG5 · 2023-11-20
**Author-Reviewer Discussion Period Ending November 22**

Hi,

Thanks for your help with the review process!

There are only two days remaining for the author-reviewer discussion (November 22nd). Please read through the authors' response to your review and comment on the extent to which it addresses your questions/concerns.

Best,\
AC